



# An analysis of the SAMA's interference on Pc3 pulsations using data from conjugate stations

Camacho Edwin[1] and Benyosef Luiz[1]

[1]COGEO, National Observatory - ON. R. Gen. J. Cristino, 77, Rio de Janeiro, 20921-400, Rj, Brazil.

**Correspondence:** Camacho Edwin (edwincmch@gmail.com)

**Abstract.** In this study, we analyzed Pc3 pulsation data from pairs of conjugate stations located at low latitudes (L-shell $< 2$). One pair of stations is situated in the Americas under the influence area of the South Atlantic Magnetic Anomaly (SAMA). As a reference, we selected a pair of stations located at a distant longitude in the Asia-Pacific region. This choice of regions facilitates meaningful comparisons. We conducted a signal spectral analysis using the Fast Fourier Transform (FFT), Continuous Wavelet
Transform (CWT), and Wavelet Coherence to characterize the dynamics of Pc3 pulsations at conjugate stations in the time-frequency domain. The Pc3 pulsations exhibited similar waveforms and occurred simultaneously at the conjugate stations in both regions. Spectral power plots and wavelet scalograms revealed notable similarities between the stations. However, we observed an increase in pulsation amplitude and wavelet coefficients at the station located near the SAMA center. Additionally, high coherence and in-phase signals were observed in the Asia-Pacific region, while moderate to high coherence signals were
observed at the station in the SAMA region. We suggest that the observed differences at the SAMA station may result from the unique characteristics of the region. The presence of the SAMA facilitates the precipitation of energetic particles from the inner radiation belt, thereby enhancing ionospheric conductivity.

## 1 Introduction

The South Atlantic Magnetic Anomaly (SAMA) is the most significant and largest Earth's magnetic anomaly, a region that
spans from the Pacific Ocean over southwestern Africa, covering latitudes between $15°$ N and $55°$ S (Nasuddin et al., 2019; Pavón-Carrasco and De Santis, 2016; Hagen and Azevedo, 2024). In this region, the main magnetic field experiences a substantial reduction, decreasing by about 22,000 nT, which is roughly one-third of its maximum value, compared to other longitudes (Caraballo, 2016; Domingos et al., 2017; Nasuddin et al., 2019). Sanchez et al. (2020) identified the current minimum value of the South Atlantic Magnetic Anomaly (SAMA) in northern Argentina, noting that its center has historically shifted from
Southern Africa to South America over the past 300 years (Domingos et al., 2017; Hartmann and Pacca, 2009; Pavón-Carrasco and De Santis, 2016). Due to the reduced magnetic field intensity over the South Atlantic Magnetic Anomaly (SAMA) region, energetic particles trapped in the inner radiation belt and drifting azimuthally undergo deep precipitation into the ionosphere and atmosphere, thereby creating a region of high radiation (Abdu et al., 2022; Nasuddin et al., 2019). The particle precipitation results in the enhanced ionospheric conductivity at the altitude of $D$ and $E$ layers due to an increase of ionization during both
the quiet and disturbed periods, in this region compared to other places, and suggested by observations of the ionosondes, ra-





diometer, and very low frequency (VLF) radio propagation Trivedi et al. (2005). Abdu et al. (2022) pointed out that ionospheric conductivity is generally higher during disturbed periods compared to quiet ones. The increase in energetic particles, in this region, can impact objects orbiting Earth, such as satellites and the International Space Station (ISS). On the Earth's surface, this can lead to disruptions in communications and GPS, creation of no-fly zones, and the induction of currents in pipelines and transmission lines (Hartmann and Pacca, 2009; Caraballo, 2016; Mendes da Costa et al., 2011). Recently, a secondary minimum within the SAMA has been observed near southwestern Africa Terra-Nova et al. (2019), raising speculation that the expansion of the SAMA might signal an impending geomagnetic reversal (Finlay et al., 2020; Sanchez et al., 2020; Yue et al., 2024). Finally, remember that although this magnetic anomaly is the most significant and largest, relatively few studies have been conducted in this region compared to other parts of the world.

Magnetic pulsations are the ground manifestation of Ultra Low frequency (ULF) hydromagnetic waves propagating in the magnetosphere. Frequencies typically range between about 1 mHz and 10 Hz; ground amplitudes range from less than 0.1 nT to tens or hundreds of nT and generally increase with latitude up to auroral regions. In this work, we focus on Pc3 magnetic pulsations ($22-100$ mHz), because some studies have shown that Pc3 magnetic pulsations play a crucial role in transferring solar wind energy into the inner magnetosphere and the dynamic coupling of the magnetosphere and the ionosphere. This process is essential for comprehending key aspects of the solar wind-magnetosphere interaction (Kamide and Chian, 2007; Francia et al., 2009; Yumoto et al., 1986). Besides, Pc3 geomagnetic pulsations can be generated either externally or internally with respect to the magnetosphere (Kamide and Chian, 2007; Waters and Menk, 2013). Pc3 pulsations are primarily associated with the transfer of upstream interplanetary waves into the magnetosphere, generated by ion cyclotron instability. This instability occurs when solar wind protons are reflected from the bow shock along the interplanetary magnetic field (IMF) lines (Greenstadt et al., 1981; Ndiitwani and Sutcliffe, 2009; Kleimenova et al., 2012; Sutcliffe et al., 2013; Yumoto et al., 1986; Ponomarenko et al., 2010; Francia et al., 2009, 2012). Previous studies have suggested that Kelvin–Helmholtz instability (KHI) at the magnetopause's flanks plays a significant role in generating and amplifying Pc3 pulsations (Yagova, 2017; Yumoto, 1986; Santarelli, 2003). Furthermore, Pc3 pulsations observed at low latitudes have been attributed to cavity or waveguide mode oscillations within the plasmasphere (Yagova et al., 2017; Yumoto et al., 1986; Santarelli et al., 2003). Kivelson and Southwood (1985) proposed that the magnetospheric cavity or waveguide mode acts as a resonator for these pulsations. However, some studies suggest that the wave structure can also be influenced by cavity or waveguide modes (MENK F., 2000). Field Line Resonance (FLR) is another proposed mechanism and energy source for Pc3 pulsations observed on Earth, particularly at higher latitudes. FLRs, which resemble vibrating strings fixed at both ends, are widely accepted as the cause of Pc3 pulsations at high and middle latitudes (Kamide and Chian, 2007; Waters and Menk, 2013; Obana et al., 2005; Waters et al., 1991; Ndiitwani and Sutcliffe, 2009; Menk et al., 2006). However, some Pc3 pulsations, especially those observed at very low latitudes (($L-shell \leq 1.2$), are better explained by cavity mode oscillations in the plasmasphere (Yumoto et al., 1985, 1986; Kivelson and Southwood, 1985). Lastly, it is worth noting that while most research on Pc3 pulsations has focused on high and middle latitudes, the SAMA region remains less explored.

This study includes data from conjugate stations, where two points on Earth's surface are geomagnetically conjugate if they lie at opposite ends of the same geomagnetic field line, located in opposite hemispheres (Wescott, 1966; Nagata, 1967; Timoçin





et al., 2018). Conjugate phenomena refer to events that occur symmetrically and simultaneously in conjugate regions, facilitated by the direct connection of magnetic field lines between the hemispheres (Nagata et al., 1962; Wescott, 1961). While much of the research on conjugate points has focused on high and middle latitudes (Obana et al., 2005; Liu et al., 2003; Takasaki et al., 2008; Shi et al., 2020; Nagata, 1967; Wescott, 1961; Nagata et al., 1962; Wescott and Mather, 1965a, b; Hajkowicz,

2006), studies at low latitudes remain less common (Yumoto et al., 1985; Ndiitwani and Sutcliffe, 2009; Pilipenko et al., 2008; Feng et al., 1995). Few studies have examined conjugate station data at low latitudes, with notable examples including Feng et al. (1995); Wolfe et al. (1990); Yumoto et al. (1985); Takahashi et al. (1994), which investigated the characteristics of Pc3 magnetic pulsations. Finally, examining Pc3 pulsation characteristics using data from geomagnetically conjugate stations offers valuable insights into the role of the ionosphere and plasmasphere in the energy transfer and propagation of these pulsations

(Waters et al., 1991; Obana et al., 2005; Yagova et al., 2017).

The aim of this work is to compare Pc3 pulsation patterns between conjugate stations, investigate their electrodynamic behavior, and evaluate the influence of the SAMA on these pulsations. The motivation for this study is to address the following fundamental question: *Are there differences in the characteristics of Pc3 pulsations recorded at conjugate stations influenced by SAMA?* Magnetic data were simultaneously recorded at two conjugate station pairs: one in the America-SAMA region and the

other in the Asia-Pacific region, allowing for a comparative analysis of Pc3 pulsation characteristics. Data were collected near the equinox period to ensure a better condition of interhemispheric symmetry. To characterize the behavior of Pc3 pulsations at conjugate stations, we employ a three-step signal analysis approach. First, we applied the Fast Fourier Transform (FFT) with Welch's power spectral method to provide a global overview of the dataset, highlighting the frequency characteristics of the pulsations between conjugate stations. This helps us to understand how the pulsations are generated and propagated. Second,

the Continuous Wavelet Transform (CWT) is used to observe time-scale variability and non-stationary features. Finally, wavelet coherence analysis is used to identify consistent scales and assess local similarities in Pc3 pulsations between stations. To our knowledge, this integrated methodology has not previously been applied to study Pc3 pulsations and their relationship with the SAMA.

The structure of this work is as follows: Section 2 presents the data, Section 3 describes the signal analysis methodology,

Section 4 presents and discusses the results, and Section 5 provides the conclusions.

## 2 Data

In this section, we provide an overview of the dataset used for this investigation, detail the ground-based stations considered, and describe the space environment conditions relevant to the case study.

### 2.1 Geomagnetic Dataset

In this work, we use the geomagnetic horizontal "$H$" component, which exhibits a maximum amplitude and is more susceptible to magnetosphere-ionosphere modulation effects. Consequently, it is more suitable for studying pulsations in conjugate points





(Obana et al., 2005; Tanaka et al., 2004; Wescott, 1961). The data were sampled at a rate of 1 second for all stations, allowing us to effectively analyze Pc3 pulsations.

The data used in this study corresponds to the period between 01:00 UT and 04:00 UT on October 25th, 2016, which corresponds to a quiet to moderately disturbed geomagnetic environment (3.5 < Kp < 5; Dst $\sim -30$ nT). More information about the space environment conditions for the selected data period is presented and discussed in Camacho et al. (2023). Previous studies have shown that some Pc3 pulsations occur in the range of $1 < Kp < 3.5$, during quiet to disturbed periods (Feng et al., 1995; Da Silva et al., 2020; Motoba et al., 2017; Yagova et al., 2015). A storm sudden commencement (SSC) was recorded at 09:22 UT on the same day, after the selected period. This time period was chosen for the analysis of Pc3 pulsations because it is the only period with data from all four stations, where the amplitudes exceed 0.5 nT, and the wave packet contains at least three wave cycles.

Fortunately, the chosen period occurred near the September equinox, ensuring nearly equal solar illumination of both hemispheres. This condition results in inter-hemispheric symmetry, with similarities in conductivity and atmospheric electric current systems at conjugate points (Hartinger et al., 2017; Timoçin et al., 2018). Such atmospheric characteristics simplify the physical system, as conjugate ionospheres during winter or summer would exhibit significantly different conductivities (Menk et al., 2006). Additionally, the moderately disturbed period, characterized by variations in solar wind pressure, offers a unique opportunity to investigate the electrodynamical coupling between the solar wind and the magnetosphere-ionosphere system at low latitudes.

## 2.2 Ground-Based stations

The selected stations are located at low latitudes ($L - shell < 2$) and are near conjugate points on the same magnetic field line. Specifically, the Kakioka station (KAK) in Japan is roughly conjugate to Alice Springs station (ASP) in Australia, while San Juan station (SJG) in the United States is roughly conjugate to São Martinho da Serra station (SMS) in Brazil. Figure 1 illustrates the locations of the conjugate stations. Additionally, the surface area of the SAMA is highlighted on the map by the closed blue line, which corresponds to the 32,000 nT magnitude isoline at the Earth's surface, as calculated using the IGRF-13 model.

The conjugate coordinates were determined using the Altitude Adjusted Corrected Geomagnetic (AACGM) model, which utilizes the IGRF-13 magnetic field model to trace magnetic field lines between hemispheres. The magnetic field line is numerically traced from the geographic starting position to the magnetic dipole equator, which is defined by the best-fit Earth-centered dipole. The AACGM latitude $\lambda_m$ and longitude $\phi_m$ are then determined by the latitude and longitude of the dipole magnetic field line that connects this point to the Earth's surface. The dipole latitude can be determined from the $L$-shell of the intersection point on the magnetic equator (Shepherd, 2014; Laundal and Richmond, 2017). While the model identifies some "forbidden regions" near the magnetic equator where AACGM coordinates cannot be defined, recent updates to the AACGM and IGRF models have reduced the size of these regions. We verified that no stations in this study fall within these forbidden zones using the Python wrapper tool[1]. The AACGM model also provides the maximum altitude ($h_{eq}$) of the magnetic flux line

---

[1]https://pypi.org/project/aacgmv2/





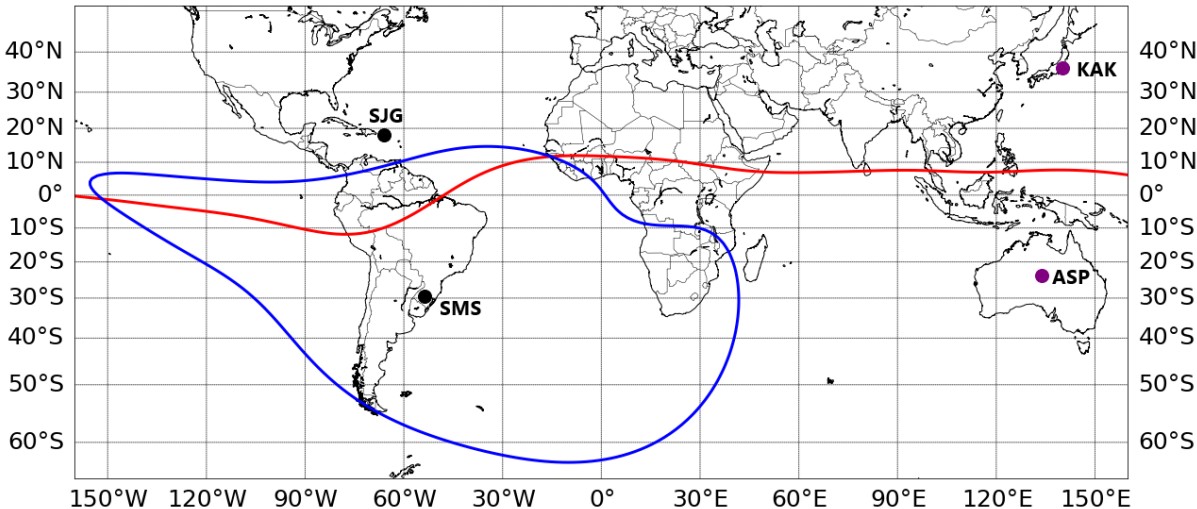

**Figure 1.** The map shows the locations of the conjugate stations, indicated by circles. The red line represents the dip magnetic equator, while the blue closed line surrounds the SAMA region.

linking the stations. All stations are part of the INTERMAGNET and EMBRACE MagNet networks (Denardini et al., 2018). Table 1 lists the geographic and conjugate coordinates, magnetic field intensity ($B$), and $L$-shell values for each station.

| | | | Coordinates | | | | | | |
| | | | Geographic | | Conjugate | | | | |
| **Stations** | **Code** | **UN**[a] | **Lat** [°] | **Lon** [°] | **Lat** [°] | **Lon** [°] | **B [nT]** | **h**$_{eq}$ **[km]** | **L** [$R_E$] |
| Kakioka | KAK | JP | 36.23 | 140.18 | -20.21 | 138.97 | 46737 | 2417 | 1.47 |
| Alice Springs | ASP | AU | -23.77 | 133.88 | 39.93 | 138.97 | 53139 | | |
| San Juan (Puerto Rico) | SJG | US | 18.11 | -66.65 | -35.20 | -55.06 | 37284 | 1260 | 1.28 |
| São Martinho da Serra | SMS | BR | -29.44 | -53.82 | 14.38 | -64.40 | 22460 | | |

UN[a] = Country, JP = Japan, AU=Australia, US= The Unided States, BR=Brazil.

**Table 1.** Coordinates of the magnetic stations.

## 3  Methodology

The spectral analysis of Pc3 pulsation variations involved several steps. Initially, the raw geomagnetic H-component data were band-pass filtered using an infinite impulse response (IIR) filter (Shaofeng, 2000; Haykin and Van Veen, 1998; Stearns and

Hush, 2016), with a unit response in the period range of 22-100 mHz, in order to extract Pc3 signals.





## 3.1 Spectral Analysis

In this work, we applied two techniques for spectral analysis: Time-Frequency analysis, which utilizes the Fourier transform, and Time-Scale analysis, based on the continuous wavelet transform.

### 3.1.1 Time-Frequency analysis

The Fourier transform decomposes a waveform into a sum of sinusoids with varying frequencies. It essentially represents the same information as the original waveform, but in the frequency domain instead of the time domain. The Fourier transform is commonly used for stationary signal analysis, where all frequencies exhibit infinite coherence (Domingues et al., 2005). This spectral analysis provides a comprehensive description of the frequency components within a given time series. The Fast Fourier Transform (FFT) is an optimized algorithm designed to compute the Fourier Transform (FT) more efficiently. The

equation below represents the continuous-time Fourier transform (Brigham, 1988; Kumar and Foufoula-Georgiou, 1997).

$$F(\xi) = \int_0^T f(t) e^{-2\pi \iota \xi t} \, \mathrm{d}t, \tag{1}$$

where $T$ is the record length, $\xi$ is the frequency, and $f(t)$ is the time series from which we extract a record of length $T$, in this case, a Pc3 pulsation time series.

In this analysis, we used Welch's power spectral density method with Fast Fourier Transform (FFT) to eliminate low-

frequency variations, allowing the spectral features within the frequency range of interest to be more clearly identified. A Hanning window with a length of 512 seconds was applied to the data in the time domain.

### 3.1.2 Time-Scale analysis

Our time-scale analysis involves calculating wavelet coefficients using the continuous wavelet transform (CWT) of the filtered signal ($f(t)$, Pc3 pulsations). The CWT is a powerful tool for analyzing both stationary and non-stationary time series, in-

cluding geophysical signals (Addison, 2002; Torrence and Compo, 1998). It provides insights into the central frequencies (the inverse of central periods) of the events as well as their timing. The wavelet coefficients $\mathcal{W}_f^\psi(a, \tau)$ generated by the CWT are defined as follows:

$$\mathcal{W}_f^\psi(a, \tau) = \frac{1}{\sqrt{a}} \int_{-\infty}^{\infty} f(t) \overline{\psi(\frac{t-\tau}{a})} \, \mathrm{d}t \qquad a > 0, \tag{2}$$

Where "$a$" is scale, "$\tau$" denotes translation and $\psi_0$ is the analyzing wavelet function, and is defined on the open time and

real scale ($a$, $\tau$) half-plane (Daubechies, 1992; Antoine et al., 2004) The analyzing wavelet function used in this work is the Morlet function defined by:





$$\psi(t) = \pi^{-\frac{1}{4}} \, \exp(\imath\,5\,t) \exp\left(-\frac{t^2}{2}\right). \tag{3}$$

where "$\omega = \imath\,5\,t$" is a non-dimensional frequency parameter of the Morlet function, chosen to meet the wavelet admissibility condition, ensuring that the integral of the analyzing wavelet is nearly zero for this value. Our goal is to distinguish the
distribution of signal energy across different scales over time. This distribution is visualized in a scalogram, which highlights the maximum energy values and their temporal occurrences. In essence, the scalogram represents the distribution of signal energy with respect to both time "$\tau$" and scale "$a$".

### 3.1.3 Global Wavelet Spectrum

The Global Wavelet Spectrum (GWS) is a mathematical tool that provides an unbiased and consistent estimation of the true
power spectrum of a time-series signal. By using wavelet transforms, the GWS evaluates how the energy (or power) of the signal is distributed across different time scales, offering both temporal and frequency information. This method is particularly useful for analyzing non-stationary signals, such as geomagnetic pulsations (e.g., Pc3), where the signal characteristics change over time. Although the GWS can be considered a smoothed version of the Fourier spectrum, it has key advantages, such as its ability to clearly identify bursts or sudden changes at high frequencies. This is because the GWS does not assume
stationarity (Daubechies, 1992; Torrence and Compo, 1998; Frick et al., 1998). This energy is typically associated with the wavelet coefficients as:

$$E_f^{\psi}(a,\tau) = \left|\mathcal{W}_f^{\psi}\right|^2 \tag{4}$$

### 3.2 Coherence Analysis

Coherence analysis can quantify the correlation between signals or similarity between them in the frequency domain (Bortel
and Sovka, 2007), helping to determine whether pulsations come from the same source region, in the frequency domain (Jun et al., 2014, 2016). Differences in observations may indicate disturbances from local processes or asymmetrical effects from external sources. The cross-wavelet transform is a tool used to examine time-scale dependencies between two time series, defined as follows:

$$\mathcal{W}_{f,g}^{\psi}(a,\tau) = \mathcal{W}_f^{\psi}(a,\tau) \, \mathcal{W}_g^{\psi}(a,\tau), \tag{5}$$

Based on this cross-transform, we can compute the wavelet coherence and a phase measure. They provide linear-quantitative estimators of the degree the relationship by scales between the two signals (Labat, 2005; Torrence and Compo, 1998). Here, the wavelet coherence is computed using the following expression:



$$\mathcal{C}_{\mathcal{W}}^2(a,\tau) = \frac{|S(a^{-1}\mathcal{W}_{f,g}^{\psi}(a,\tau))|^2}{S(a^{-1}|\mathcal{W}_f^{\psi}(a,\tau)|^2)\,S(a^{-1}|\mathcal{W}_g^{\psi}(a,\tau)|^2)} \tag{6}$$

Wavelet coherence ($\mathcal{C}_{\mathcal{W}}^2(a,\tau)$) values, ranging from 0 to 1, indicate the degree of linear relationship between two signals.
Values close to 1 suggest a strong correlation, implying the signals likely come from the same source. Values near 0 indicate weak or no correlation, suggesting the signals originate from different sources. The operator "$S$" is a smooth operator in the time-scale domain, implemented as a convolution of transform coefficients with a Gaussian moving average for smoothing in both time and scale directions (Grinsted et al., 2004). This smoothing helps reduce noise caused by amplification provided by the product. The resulting coefficients are visualized in a color-map graph, similar to a scalogram in wavelet transforms. The
wavelet coherence phase, denoted as $\theta_{x,y}$, is derived from the real and imaginary components of the coefficients.

$$\theta = \tan^{-1}\left(\frac{\Im\left\{S(a^{-1}\mathcal{W}_{f,g}^{\psi}(a,\tau))\right\}}{\Re\left\{S(a^{-1}\mathcal{W}_{f,g}^{\psi}(a,\tau))\right\}}\right) \tag{7}$$

This measurement yields information on the delay of one signal considering the other as a function of time and scale. Implementation of the CTW and wavelet coherence were based on a library of MATLAB functions provided by Grinsted et al. (2004).

## 4   Results and Discussions

Our study focuses on the spectral analysis of Pc3 pulsations from terrestrial magnetic stations located at conjugate points in two global regions. The objectives include identifying Pc3 events, analyzing power spectral density, performing wavelet analysis to examine process dynamics, conducting coherence analysis between conjugate stations, and comparing behaviors influenced by the SAMA.

### 4.1   Pc3-Event Identifications

Figure 2 shows the Pc3 pulsations for each conjugate pair (with each pair indicated by red and blue). We initially selected two time intervals representing simultaneous wave packets of Pc3 pulsations. Both intervals have the same duration of 10 minutes. The time intervals were chosen based on the presence of significant Pc3-pulsation amplitudes in both regions and the significant wavelet coefficients (as seen in Figs. 4 and 5, which are described later).
Through visual inspection, the plots in Fig.2 show consistent propagation patterns of Pc3 wave packets that are simultaneous and similar between the conjugate stations, particularly between KAK and ASP in the Asia-Pacific region. That is, the occurrence of the Pc3 pulsation was essentially identical, and relative amplitudes were similar. This observation is consistent with some previous investigations at conjugate stations (Obana et al., 2005; Feng et al., 1995; Takahashi et al., 1994; Engebretson et al., 2000). Additionally, the panel at the bottom (b) of Fig. 2 shows that Pc3 wave packets in the southern hemisphere




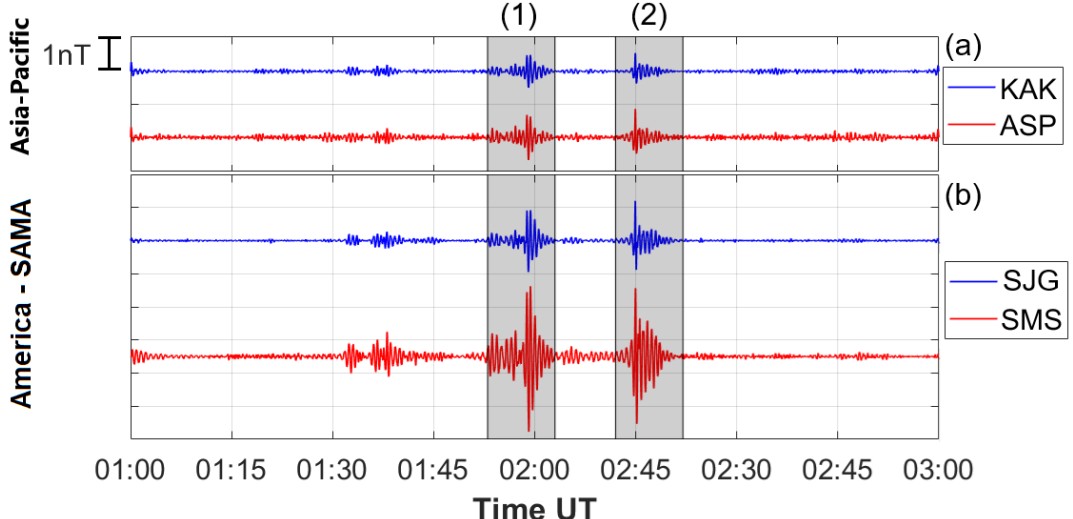

**Figure 2.** Filtered Pc3 pulsations at conjugate stations, on October 25th, 2026. (a) KAK-ASP and (b) SJG-SMS.

generally exhibit higher amplitude intensities compared to those in the northern hemisphere for conjugate station pairs. At the SMS station, an enhancement in Pc3 pulsation amplitudes is observed near the center of the South Atlantic Magnetic Anomaly (SAMA). During quiet to moderately disturbed geomagnetic conditions, the SAMA region experiences an increase in Pc3 pulsation amplitudes, which may be associated with electron precipitation in the ionospheric $D$ and $E$ layers. The SMS station, located near the center of the SAMA, recorded the highest amplitudes of these pulsations.

**4.2 Pc3 Power-Spectral density analysis**

We employ a power spectral method (explained in section 3.1.1) to analyze the significant peaks and evaluate the similarity between the conjugate station pairs. Figure 3 displays Welch's power spectral density (PSD) of the filtered time series, calculated using the FFT, for the selected intervals. It also shows the frequency contributions to the Pc3 pulsations across all intervals for the conjugate pairs. In interval 1, two significant discrete spectral peaks are observed for each conjugate pair at 23.4 mHz and

29.2 mHz in both regions. In interval 2, a significant peak is observed at 27.3 mHz in all stations. Furthermore, the amplitudes of the peaks in both intervals are similar in the Asia-Pacific region. In contrast, the amplitudes of the peaks at the SMS station are considerably higher than those at its conjugate station. This result may indicate that the SMS station is influenced by an altered upper atmosphere in the SAMA region, primarily due to the ionosphere. It is important to note that some authors have observed spectra similar to Pc3 pulsations at low latitudes and/or over a wide range of latitudes on the Earth's surface (Yumoto

and Saito, 1983; Odera et al., 1994; Matsuoka et al., 1997). However, this would be the first time that similar spectra have been observed at conjugate points and at low latitudes, particularly when considering the SAMA region.

The dominant frequencies (23.4, 29.2, and 27.3 mHz) are consistent with some of the discrete frequencies observed in the 20–30 mHz range, as reported by (Menk et al., 1994; MENK F., 2000; Ndiitwani and Sutcliffe, 2009; Eriksson et al.,





2006; Feng et al., 1995). Several studies have reported the manifestation of pulsations and fluctuations at discrete frequencies, often interpreted in terms of field line resonances (FLRs) associated with large-scale waveguide/cavity modes in the Earth's magnetosphere (Kamide and Chian, 2007; Waters and Menk, 2013; MENK F., 2000; Ndiitwani and Sutcliffe, 2009; Villante et al., 2022; Waters et al., 2000; McPherron, 2005). However, this topic is not the primary focus of the current work, though it could be explored in future studies.

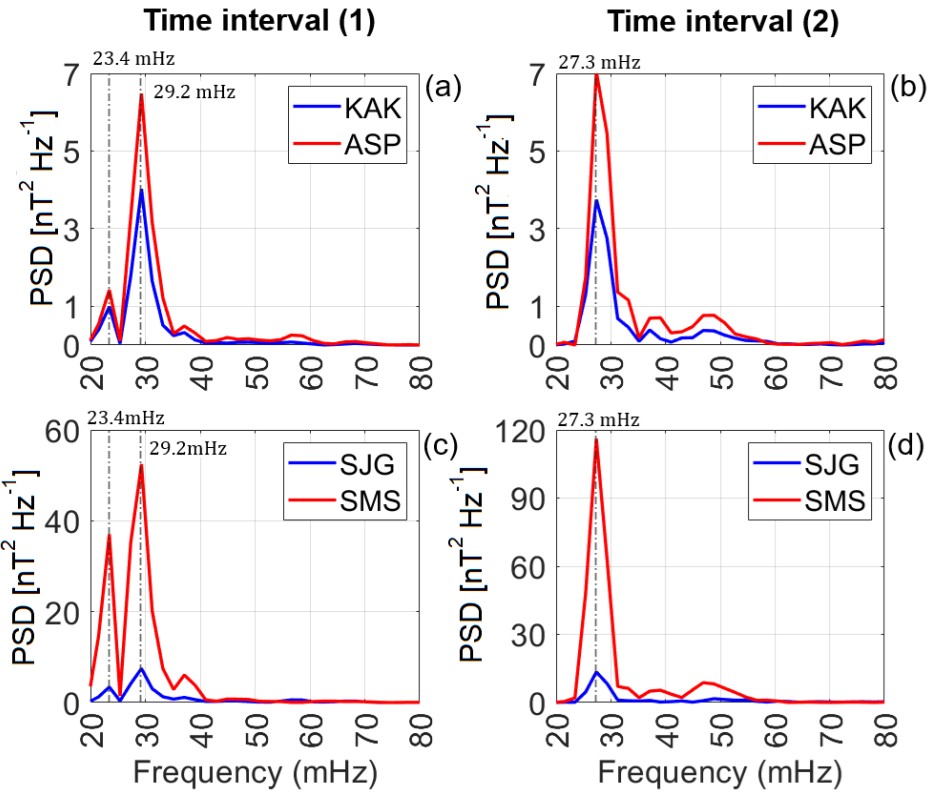

**Figure 3.** Welch's power spectral density of the Pc3 pulsations from conjugate station in the Asia-Pacific region: time interval 1 (a) and (c), time interval 2 (b) and (d).

## 4.3 Wavelet analysis of Pc3 dynamics

In this section, we analyze the signal energy across different scales over time. Scalograms are used to visualize the location of maximum values and their temporal occurrence, offering an effective way to examine pulsation details and highlight similarities.

For the Asia-Pacific region, Fig. 4 shows the Pc3 pulsation signal (top panels "a" and "b"), the corresponding scalograms (middle panels "c" and "d"), and plots of the global wavelet spectrum (bottom panels "e" and "f"), each corresponding to the conjugate stations. Panels in the same vertical column correspond to the KAK station in the northern hemisphere (on the left)





and the ASP station in the southern hemisphere (on the right), while panels at the same horizontal position refer to the conjugate station pair. In wavelet scalograms, the horizontal axis represents the time in hours, while the vertical axis represents the signal intensity (in $nT$) and wavelet scale (in $mHz$), respectively. The magnitudes of the wavelet coefficients are indicated by the color bar, which has the same range ($nT^2\ mHz^{-1}$) for the signal analysis of each station.

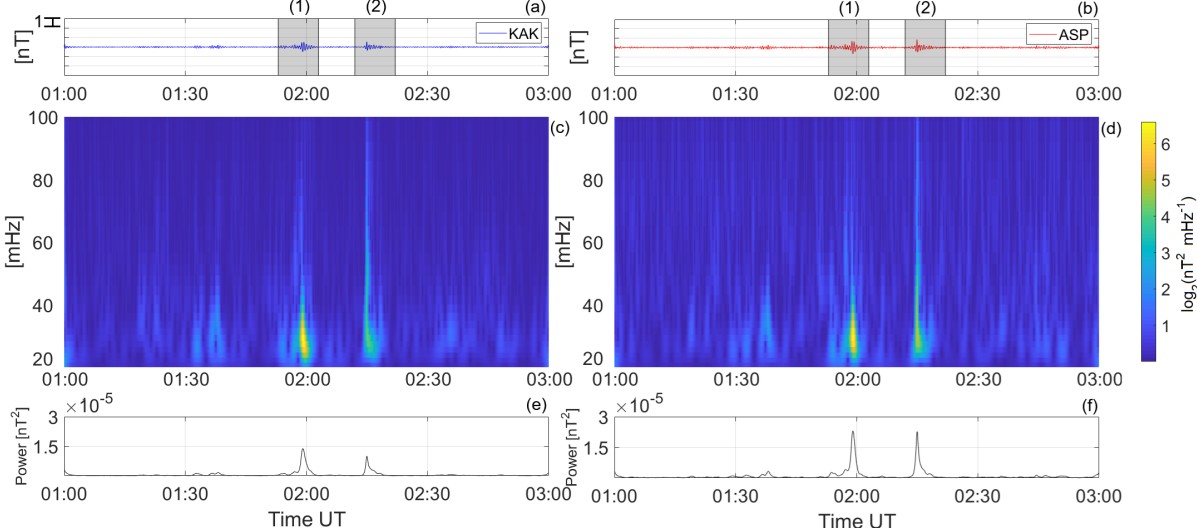

**Figure 4.** Pc3 pulsations (a-b), squared-wavelet coefficients scalograms (c-d) and global wavelet spectral (e-f). Asia-Pacific region.

Figure 4, panels "c" and "d", highlights the characteristics of the Pc3 pulsations. The maximum values of the wavelet coefficients (indicated by the yellow color) coincide with the most intense Pc3 wave packets observed at each station (panels "a" and "b"). In the two selected time intervals (1 and 2), the scalograms show higher intensity spectrum values in the frequency range of 20 to 40 mHz. An inspection of the scalograms reveals similar features in energy amplitude, duration, and scale (converted to central frequency) for the conjugate pairs. The global wavelet spectra plots highlight two peaks that are similar

between the conjugate stations, coinciding with the Pc3 pulsation wave packets observed at each station. However, it can be noted that the peak powers at the ASP station are slightly more intense. Based on the observed similarity in energy patterns, we conclude that the conjugate pairs in the Asia-Pacific region are subject to nearly identical conditions in the magnetosphere-ionosphere system.

     Figure 5 presents the Pc3 pulsation data (top panels), scalogram (middle panels), and plots of the global wavelet spectrum

(bottom panels) for the conjugate station, following a format similar to the previous figure, but focusing on the America-SAMA region.

     Initially, the scalograms show similar and simultaneous energy patterns, with the more intense wavelet coefficients corresponding to the Pc3 pulsation wave packets. However, the coefficients at the SMS station show a slight increase in intensity, particularly during the time interval (2). Additionally, the more intense wavelet coefficient values in the scalograms are ob-

served between 20 and 40 mHz. The global wavelet spectra plots (panels "e" and "f") reveal two peaks at the conjugate stations,



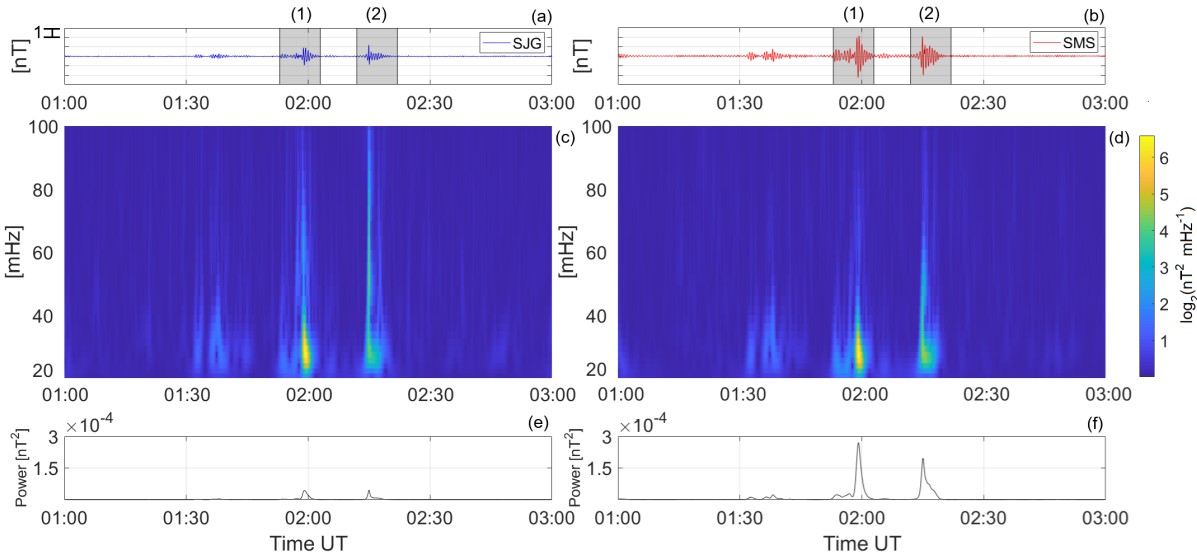

**Figure 5.** Pc3 pulsations (a-b), squared-wavelet coefficients scalograms (c-d) and global wavelet spectral (e-f). America-SAMA region.

which align with the Pc3 pulsation wave packets detected at each location. However, it is noted that the peak powers at the SMS station are more intense than at its conjugate. Finally, the results suggest that the SAMA influences the energy of Pc3 pulsations due to ionization in the SAMA region. The use of the CTW technique to investigate time-scale characteristics has significantly contributed to this study.

### 4.4 Coherence Analysis

To complete the investigation, we employ coherence analysis to address the following question: For a pair of signals from conjugate stations, how similar are their power spectra?

The wavelet coherence for each conjugate pair in both global regions, calculated using Eq. (6), is used to create colormap plots (Fig. 6). These plots represent the coherence values of Pc3 pulsations between the conjugate pairs, with yellow indicating maximum coherence (close to one). Arrows indicate the phase relationship between signals: right arrows for in-phase, left arrows for anti-phase, and inclined arrows for intermediate values. Arrows are shown only where the coherence is 0.8 or higher for better visualization.

Figure 6 shows the wavelet coherence results for the conjugate stations in both the Pacific-Asia and America-SAMA regions. As seen in Fig. 6, panel "a," the time intervals for the KAK-ASP stations exhibit very high coherence ($\mathcal{C}_\mathcal{W} > 0.7$) and in-phase signals for pulsations at the conjugate stations. Coherence in the 20 to 100 mHz band is consistently high between this conjugate pair. Matsuoka et al. (1997) and Obana et al. (2005) also observed high-coherence Pc 3–5 pulsations (using FFT) between conjugate stations, recorded in the Asia-Pacific region. Finally, the high coherence in this region suggests that the Pc3 pulsations observed at the conjugate stations likely originated from the same source in the magnetosphere. These



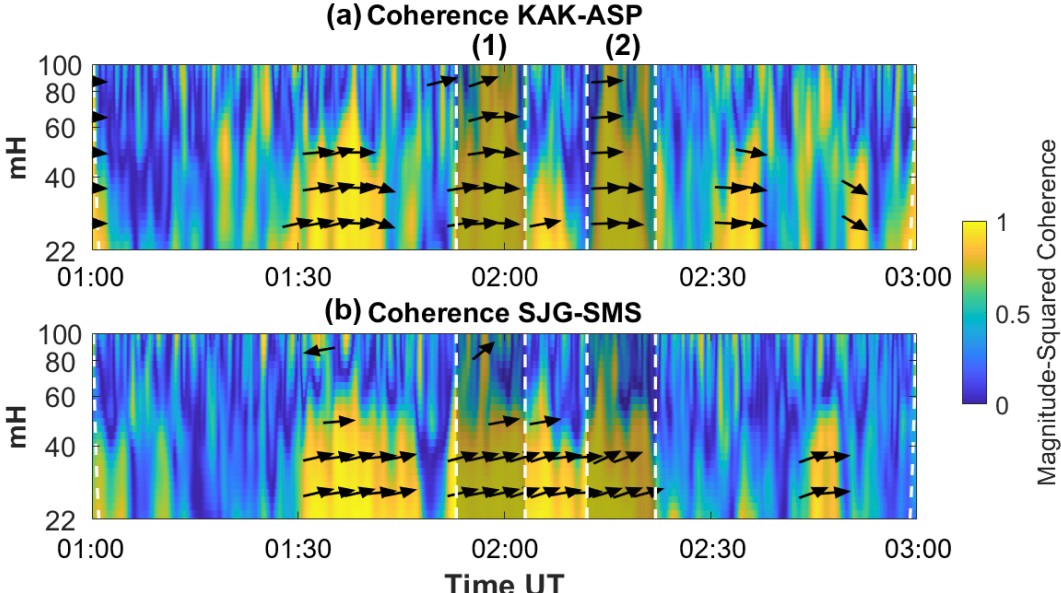

**Figure 6.** Wavelet coherence representations for Pc3 pulsations between the conjugate pair stations. Color map indicates the amplitude, and arrows the phase where the amplitude is larger than 0.8. (a) KAK-ASP and (b) SJG-SMS.

results may indicate that the stations are subject to similar effects and undergo similar processes of energy transfer between the magnetosphere and ionosphere.

In contrast, when examining the time intervals in Fig.6b, there are generally moderate to high signal coherence ($0.4 < \mathcal{C}_\mathcal{W} < 0.7$) and in-phase signals between the conjugate stations. The SJG-SMS pair shows lower coherence than the KAK-ASP pair, with moderate to high coherence observed in the 20 to 60 mHz band. This reduced coherence for SJG-SMS may be linked to the station's proximity to the SAMA center, where enhanced electrical conductivity, due to particle precipitation, alters pulsations and impacts the plasmasphere-ionosphere system, thereby modifying the structure of Pc3 pulsations along the magnetic field lines connecting conjugate stations.

### 4.5 Summary of the comparative analysis of PC3 pulsations inside and outside SAMA

In summary, striking similarity and simultaneity of Pc3 pulsations were observed at conjugate stations at low latitudes, and in two distinct regions. The pulsations occurred near the period of interhemispheric illumination symmetry, coinciding with the September equinox.

In the Asia-Pacific region, nearly symmetric conditions at the magnetopause, plasmasphere, and ionosphere result in similar and simultaneous Pc3 pulsation amplitudes. Spectral analyses (PSD) and scalograms show significant similarities between conjugate stations, with high coherence and in-phase signals in this region.





In contrast, in the America-SAMA region, an enhancement in the amplitude of Pc3 pulsations was observed at the SMS
station. While spectral analyses and scalograms showed similarities, the SMS station displayed more intense power spectral
density and wavelet coefficients. Notably, the discrete spectral frequencies (PSD) were identical in both regions. Moderate to
high coherence and in-phase signals were also observed in this region. These differences arise from the asymmetric condi-
tions between the America-SAMA stations. Although both stations experience similar magnetopause conditions, they differ
in particle precipitation and plasmasphere-ionosphere dynamics, with the SMS station exhibiting an ionization anomaly. This
suggests that the SAMA region influences the characteristics and structure of Pc3 pulsations.

The high coherence observed in both regions suggests that the Pc3 pulsations likely originate from the same external source
in the magnetosphere, propagating along magnetic flux lines linking the conjugate stations. However, at the SMS station, the
Pc3 pulsations may also be influenced by factors such as the ionization in the SAMA, particle precipitation, and variations in
electrical conductivity during slightly perturbed periods. These asymmetries in plasmasphere-ionosphere conditions alter the
Pc3 pulsations along the magnetic flux lines in the Americas-SAMA region, contrasting with the Asia-Pacific region, which is
under more symmetric conditions. These findings support the conclusion that the SAMA region influences the characteristics
of Pc3 pulsations. This study does not address the sources of the pulsations or the internal mechanisms driving the modifi-
cations in the magnetic pulsations. Exploring these aspects would necessitate the use of additional methodologies, as well as
comprehensive data from ground-based instruments to examine the ionosphere and from satellites to probe the plasmasphere.

## 5  Conclusions

This study analyzed Pc3 pulsation data from two conjugate station pairs at low latitudes ($L - shell < 2$), located in the Asia-
Pacific and American-SAMA regions. The aim was to compare the Pc3 pulsation patterns between these stations, investigate
their electrodynamic behavior at conjugate locations, and assess the influence of the SAMA region. Fourier and wavelet trans-
form techniques were applied to analyze the pulsations, allowing for the visualization of spectral and coherence characteristics.
To simplify the geophysical and electromagnetic processes in the magnetosphere-ionosphere system, data were analyzed dur-
ing a period of near interhemispherical illumination symmetry. The key findings from the Pc3 pulsation data analysis are
summarized as follows:

1. The Pc3 pulsations exhibited similar waveforms and occurred simultaneously at conjugate stations in both regions. Under
   the influence of the SAMA, these pulsations showed an enhancement in amplitude.

2. The dominant frequencies identified (1.9, 2.1, 2.6, and 2.8 mHz) were identical in both regions. However, the SMS sta-
   tion, located near the SAMA center, exhibited a general amplification of spectral power (PSD) compared to its conjugate
   station.

3. A high degree of similarity in the scalograms was observed at all conjugate stations in both regions. However, at the
   station influenced by SAMA, the wavelet coefficients were more intense.

4. In the Asia-Pacific region, the signals between conjugate pairs exhibited high coherence and in-phase behavior. In contrast, in the America-SAMA region, the Pc3 pulsations showed moderate to high coherence, also remaining in phase.

In summary, while the Pc3 wave structures likely originate from the same external source, possibly within the magnetosphere, and propagate to lower latitudes via magnetic flux lines to the conjugate stations, certain characteristics of the pulsations are modified in the SAMA influence region. The observed differences in this region, such as the enhanced amplitudes of Pc3 pulsations and more pronounced wavelet and PSD coefficients, may be attributed to the unique features of this sector. These include significant longitudinal variation in the magnetic field and the precipitation of energetic particles from the inner Van Allen belts, which alter the ionospheric conductivity.

These represent the first direct observations of Pc3 pulsations at conjugate points in the SAMA influenced region, as well as the first comparative analysis between conjugate stations, highlighting how the electrodynamics of the pulsations are influenced by the largest anomaly of the Earth. This study provides new insights into the effects of the South Atlantic Magnetic Anomaly on geomagnetic pulsations recorded at ground stations located at conjugate points. A valuable direction for future research is to investigate the role of internal factors, such as Field Line Resonance and cavity/waveguide modes, in influencing magnetic pulsations, an objective that lies beyond the scope of the present study.

*Code availability.* The software code used in this study was developed in-house and is based on predefined functions in MATLAB.

*Data availability.* Data are available from the international repositories (ex. Intermagnet) and from INPE Embrace's site.

*Author contributions.* All authors cooperate in the same proportion in the work.

*Competing interests.* The authors declare no competing interests.

*Acknowledgements.* This study was financed by the Conselho Nacional de Desenvolvimento Científico e Tecnológico (CNPq), PCI/ON/M-CTI grant 301114/2024-2. The authors would like to thank INTERMAGNET (https://www.intermagnet.org) and EMBRACE MagNet (http://www2.inpe.br/climaespacial/portal/en/) for the datasets used in this work. The authors also thank Dr. Mendes and Dra. Domingues (INPE) for their collaboration on this work.



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
