# Peer review of "Analysis of SAMA interference on Pc3 pulsations using data from conjugate stations: A case study"

_EGUsphere, 2025_

## Author Response (AR2)

**Response to Editor**

Title: **An analysis of the SAMA's interference on Pc3 pulsations using data from conjugate stations**

Reference: **egusphere-2025-794**

Author(s): **Edwin Camacho and Luiz Benyosef**

MS type: **Research article**

Special issue: **Geomagnetic observatories, their data, and the application of their data**

Dear Editor,

We have carefully addressed all comments from the reviewers and the editor, both major and minor, and have thoroughly revised the manuscript to enhance its overall quality and clarity. We sincerely appreciate the reviewers' thoughtful feedback and the valuable contributions they have made to our work. Additionally, we have made several minor textual changes as requested by the editor, which are detailed at the end of this document.

In consideration of the reviewers' comments, we have made a slight revision to the title of the article:

*"Analysis of SAMA interference on Pc3 pulsations using data from conjugate stations: A case study"*

Below, we provide a point-by-point response to each comment. The reviewers' remarks are presented in italics, followed by our detailed responses. Additionally, we have indicated the corresponding changes in the manuscript by referencing the line numbers.

**Answers to the Reviewer #1's Evaluations:**

**Comment #1:**

*"To me, in Section 3.1, such detailed description of the basics of spectral analysis is excessive, as well as the well-known wavelet functions and their equations. I recommend some reduction of this theoretical part, so that it doesn't look like a lecture note on mathematics."*

**Response:** We agree with the comment and have revised the text to summarize the basic concepts more concisely. Specifically, we removed certain mathematical expressions and sentences that are already well established in the field. These changes have been implemented in Section 3, specifically in lines 180–181, 195–197, and 201.

**Comment #2:**

*"Also please check the period axis labels in wavelet coherence plots in Fig. 6."*

**Response:** Thank you for pointing this out. Figure 6 has been updated to accurately reflect the period values and to improve the clarity of the graphs.

**Answers to the Reviewer #2's Evaluations:**

**Comment #1:**

*"As to the first, it is of course not serious to make general statements about the SAMA's influence on pulsations based on such a short data set - there is no prove that it is representative and not just by chance that intensities at one station are larger that at some other."*

**Response:** We thank the reviewer for their valuable feedback and acknowledge the concern regarding the generalizability of our findings based on a limited dataset. In response, we have revised the Discussion and Results sections to clearly state that our analysis constitutes a case study. We will ensure that the text avoids implying broader generalizations and instead emphasizes that our conclusions are specific to this particular event. We have also updated the title to reflect this focus, ensuring that readers understand it is a case study.

Regarding the comment on the observed pulsation intensities, we respectfully note that our results are consistent with previous studies. Prior work (e.g., Obana et al., 2005; Yumoto et al., 1985; Engebretson et al., 2000) has shown similar amplitude and phase characteristics at conjugate stations outside the South Atlantic Magnetic Anomaly (SAMA). Moreover, studies focused on the SAMA region (e.g., Da Silva et al., 2020; Trivedi et al., 2005) have also documented enhanced pulsation amplitudes, supporting the validity of our observations.

We would like to emphasize that our analysis is based on 1-second resolution data from conjugate stations, an uncommon and valuable dataset in this field. While the dataset's limited scope prevents a broader statistical analysis at this stage, we agree that such a study would be important and aim to pursue it in future work, depending on project scope and data availability.

As a positive contribution, we highlight the methodological strength of our study, which integrates three complementary techniques to address our research question. We believe this approach provides new insights and adds value to the literature. Furthermore, we stress the importance of well-documented case studies in advancing scientific knowledge, particularly when they rely on rare or unique datasets (e.g., Takahashi, 1994; Trivedi, 1997; Sarafopoulos, 2005; Francia et al., 2012). In this context, we are confident that our work makes a meaningful contribution to the field.

To address this point, we have revised several parts of the manuscript. Specifically, changes were made in the following sections:

- Abstract (lines 1 and 6).
- Introduction (line 59).
- Data Section (lines 94–95).
- Results and Discussion (lines 203, 295–296, 302–309, and 322–323).
- Conclusion (lines 332–333 and 355–359).

**Comment #2:**

*"As to the latter, it is misleading to include the concept of conjugate points to such an analysis in such a simplistic way. It certainly is intriguing to investigate if an anomaly of the main magnetic field has a visible impact on some field variations, but this should be done in a way that is not self-contradictory. The problem is that the concept of conjugate points (as Authors describe correctly) is based on the footpoints of the same dipole field line. However, the essence of SAMA is that there are significant deviations from the dipole approximation in this region and field lines are not dipole-like. I am afraid that it is impossible to find footpoints of the same field line with a dipole approximation for the SAMA region and hence that the distribution of the used stations is not really relevant for the study"*

**Response:**

We appreciate the reviewer's second comment and acknowledge the concern regarding the determination of conjugate points. To calculate conjugate points, we employed the Altitude-Adjusted Corrected Geomagnetic (AACGM) coordinate system. This coordinate conversion involves tracing a magnetic field line to the dipole magnetic equatorial plane using the full-resolution International Geomagnetic Reference Field (IGRF) model, including its non-dipolar components. From that point, a dipole field line is then traced back to the Earth's surface (Shepherd, 2014; Laundal and Richmond, 2017). These steps ensure a more physically meaningful mapping between conjugate locations.

We utilized Python-based tools that implement AACGM coordinates for this purpose. While it is true that AACGM coordinates are traditionally undefined in certain regions, particularly near the magnetic equator and parts of the South Atlantic Ocean, recent updates to Shepherd's (2014) original code now provide coordinate estimates in these regions through interpolation and fitting techniques. Importantly, the software is designed to alert users when a location falls within a region where the coordinate calculation is invalid. In such cases, the code returns a NaN value, indicating that the conjugate point could not be determined.

In our study, we verified that none of the stations used were located within these "forbidden" regions. Therefore, it was possible to compute valid conjugate points, even within or near the South Atlantic Magnetic Anomaly (SAMA), and to base our analysis and discussion on those results. Finally, we consider it important to mention that an online calculator for transformations between geographic and AACGM-v2 coordinates is also available (https://sdnet.thayer.dartmouth.edu/aacgm/aacgm_calc.php). This tool allows any interested reader to independently verify that the station coordinates used in our study are appropriate for calculating geomagnetic conjugate points.

At last, we have revised and rewritten Section 2.2, *Ground-Based Stations*, of the manuscript (lines 110–143) to clarify the methodology used for conjugate point calculation, including the steps taken to ensure data validity in this challenging geomagnetic region.

**Answers to the Editor's comment:**

The authors have attended to the issues raised by Referees, Section 3 of the manuscript has been revised, there is a change of the narrative focusing on one case study to draw a conclusion, avoiding generalization of the findings. An extensive explanation was given of how the conjugate stations were selected. In the "Conclusions" section, there is an acknowledgement that one case study is not enough to support fully the current findings. And the authors state the circumstances that led them to use only one case study: the limited availability of high-resolution data at conjugate stations. They promise to conduct a comprehensive statistical analysis in future to validate their findings. There are few minor corrections to support the findings from one case study and avoid absolute statements:

1. Line 335: the words "… are modified…" to be replaced with "… might be modified…".

2. Line 340: the words " … are influenced …" to be replaced with " may be influenced…".

**Response:**

We appreciate the editor's suggestions and have incorporated the recommended changes accordingly.